# Effects of Low-Volume High-Intensity Interval Exercise on 24 h Movement Behaviors in Inactive Female University Students

**DOI:** 10.3390/ijerph19127177

**Published:** 2022-06-11

**Authors:** Yining Lu, Huw D. Wiltshire, Julien S. Baker, Qiaojun Wang

**Affiliations:** 1Faculty of Sport Science, Ningbo University, Ningbo 315000, China; st20184530@outlook.cardiffmet.ac.uk; 2Cardiff School of Sport and Health Sciences, Cardiff Metropolitan University, Cardiff CF5 2YB, UK; hwiltshire@cardiffmet.ac.uk; 3Centre for Health and Exercise Science Research, Department of Sport, Physical Education and Health, Hong Kong Baptist University, Kowloon Tong, Hong Kong; jsbaker@hkbu.edu.hk

**Keywords:** movement behavior, compensatory effect, high-intensity interval exercise, Tabata, inactive females

## Abstract

The purpose of this study was to examine if low-volume, high-intensity interval exercise (HIIE) was associated with changes in 24-h movement behaviors. A quasi-experimental study design was used. We collected accelerometry data from 21 eligible participants who consistently wore an ActiGraph for a period of two-weeks. Differences in behaviors were analyzed using a paired t-test and repeated measures analysis of variance. Regression analysis was used to explore relationships with factors that impacted changes. The results indicated a compensatory increase in sedentary time (ST) (4.4 ± 6.0%, *p* < 0.01) and a decrease in light-intensity physical activity (LPA) (−7.3 ± 16.7%, *p* < 0.05). Meanwhile, moderate-intensity physical activity (MPA), vigorous-intensity physical activity (VPA), and total physical activity (TPA) increased following exercise (*p* < 0.001). Sleep duration and prolonged sedentary time were reduced (*p* < 0.05). Exercise intensity and aerobic capacity were associated with changes in ST. The results from the study indicate that participating in a low-volume HIIE encouraged participants who were previously inactive to become more active. The observations of increases in ST may have displaced a prolonged sitting time. The decrease in sleeping time observed may be reflecting an increased sleep quality in connection with increased higher-intensity PA.

## 1. Introduction

Physical inactivity has been significantly associated with a wide range of adverse health outcomes, such as an increased risk of cardiovascular diseases, Type 2 diabetes, metabolic syndrome [1,2,3,4], as well as increased mortality [5]. Recently, the rising prevalence of physical inactivity worldwide has become a major public health concern, and more than one in four adults fail to follow the minimum recommended levels of participating in 150 min of moderate physical activity (MPA) per week [6]. Reduced physical activity (PA) and increased sedentary time (ST) are also common among students, especially when they attend university [7,8,9]. Moreover, only one of five female students meet the weekly PA recommendations [10]. ST for students includes attending classes, studying, and sitting in front of computers. This accounts for most of the usable daytime and leaves limited time for PA among university students [11,12]. Therefore, there is a need to identify an efficient and effective type of physical activity to increase the level of PA participation in this population.

Recently, the prevalence and popularity of high intensity interval exercise (HIIE) among young adults has provided an exciting type of exercise and health promotion intervention, but its efficacy varies across studies [13,14,15,16,17]. The ‘‘activitystat’’ hypothesis may be one of the potential explanations for the varied efficacy observed [18]. This hypothesis suggests that increased energy expenditure during PA must be compensated for by the conservation of energy by changing the accumulation of other movement behaviors. Such changes were defined as compensatory movement behaviors [19], and in the context of HIIE, indicate that participants compensate for increased PA during HIIE by being less active following HIIE [20]. In contrast, some studies have indicated that activity synergy occurs because participating in exercise helps individuals to stay active for the rest of the day [21,22,23].

Recently, the prevalence and popularity of HIIE among young adults have made it an exciting type of exercise and health promotion intervention, especially among females as it is promoted as a time-efficient and effective strategy to reduce fat [24]. Although the effects of HIIE on improving health outcomes has been investigated by numerous studies [14,25,26,27], few studies have examined the effects on movement behaviors following high intensity exercises. A previous study evaluated the within-day changes of sedentary behaviors after vigorous physical activity and reported an increase in subsequent sedentary time [28]. Another crossover study on overweight boys examined the behavioral changes after a single bout of moderate or vigorous exercise. The author indicated a slightly higher increase in sedentary time after vigorous exercise than moderate in the following 4 days, and furthermore, the time spent in vigorous intensity physical activity (VPA) decreased greatly after vigorous exercise [29]. While in the study conducted in adults, time spent in sedentary activities was significantly reduced during HIIE [30].

When investigating movement behaviors, it is important to note that the length of the day is limited to 24 h, suggesting an increases in the time spent in one behavior must result in a decrease in the time spent in associated behaviors. There is potent evidence for sedentary behaviors [31,32], sleep [33,34,35], and MVPA [36,37,38], and emerging evidence for LPA to be associated with health outcomes [39,40]. Moreover, the allocation of whole 24 h time is associated with health outcomes across the lifespan [41,42,43]. Therefore, collecting data on all movement behaviors, including PA performed at all intensity categories, sedentary behaviors, and sleep over a full day (24 h) may provide researchers with the best method to observe any changes in the time accumulated in other behaviors, resulting from a single or combined behavioral intervention [44]. Understanding the 24 h movement behavior changes after HIIE can improve our current knowledge in relation to VPA adaptations and responses and concurrently aid in prescribing PA and health promotion programs. However, there is a paucity of research on the effects of HIIE on 24 h movement behaviors among young women. Therefore, the main purpose of this study, therefore, was to examine changes in 24 h movement behaviors following low-volume HIIE among inactive female university students. Secondly, this study also examined the perceived exertion and fidelity to gain insight into whether such vigorous intensity exercise will be acceptable for inactive female university students. We hypothesized that ST would increase following HIIE. Furthermore, we expected to see a disparity in the magnitude of increase in ST among participants with the highest mean heart rate (HR_mean_) during exercise. We also hypothesized that the greatest increases in VPA will be observed on the exercise day.

## 2. Materials and Methods

### 2.1. Participants

Healthy female students were recruited from a university. The inclusion criteria included having an inactive lifestyle, which was defined as engaging in less than 90 min of moderate intensity activity per week for a period longer than three months [45]. Participants were required to complete a PAR-Q+ questionnaire for further eligibility screening. Participants who were severely obese (BMI ≥ 40), who had smoked in the past 6 months, took hormonal medications, or who were diagnosed with medical conditions were all considered unsuitable for high intensity exercises and were excluded. Finally, 29 eligible participants were included in this study. Participants were instructed to maintain their normal dietary and lifestyle habits throughout the intervention. The study was approved by Ningbo University ethics committee (RAGH20213744). Written informed consent was provided from all participants prior to the intervention and data collection.

### 2.2. Procedures

A quasi-experimental design was used for data collection. In this 2 week study, all participants were identified as their own controls. Previous studies recommended at least 7 consecutive days, which was required to assess habitual PA [46], and a frequency of 3 times per week was preferred for HIIE to provide health promotion [47]. Therefore, to make it comparable, the protocol in this study began with a control week during which participants were asked to maintain their PA as usual. This was followed by an intervention week where each participant was required to engage in three sessions of Tabata style HIIE, with a one day interval between each session. Because of menstrual cycle differences, the starting times (at the end of menstruation) were different between participants, with the last participant starting 20 days later than the first. On the day before the commencement of the protocol, participants were asked to attend the laboratory. During the visit, participants were equipped with an accelerometer to assess movement behaviors over the two weeks including the exercise sessions. Furthermore, they were instructed to record exercise and sleep time in individual logbooks. They were fully familiarized with exercise protocols, data collection procedures, and had completed baseline assessments prior to the two-week intervention. During the exercise day, the participants were required to attend the laboratory and complete the HIIE, which was supervised by a trained researcher. During each session, a heart rate monitor was used to record the HR_mean_ and peak heart rate (HR_peak_) during exercise, which was used as a measurement of exercise fidelity. Furthermore, Borg’s rating of perceived exertion (Borg’s RPE) was recorded after each session to assess subjective perception of effort during exercise. At the end of the two-week protocol, participants returned the accelerometer and the sleep and exercise log. Flow diagram of samples and study timeline was outlined in Figure 1.

### 2.3. Measurement

Anthropometry

At baseline, body mass was determined using a calibrated bioelectrical impedance analysis (BIA) (MC-180, TANITA CO., Dongguan, China) to the nearest 0.1 kg. Height was measured using a stadiometer (HGM-6, Shanghai, China) to the nearest 0.1 cm. Body mass index (BMI) was also calculated using standard equations. Standardized procedures were used for all subjects during all measurements.

Aerobic capacity

Maximal oxygen uptake (VO_2max_) was used to measure the aerobic capacity of participants. The modified YMCA submaximal cycle ergometer test has been used previously as a reliable measure to estimate oxygen uptake in adults (Ergoselect 100, Ergoline GmbH, Bitz, Germany) [48]. This modified test includes 2–4 stages, with each stage lasting 3 min. After resting for 10 min, the participants began to cycle at 0.5 kg (25 W; 150 kg·m/min), which increases based on stable heart rate recordings during the last 1 min in stage 1. Participants were then asked to maintain a steady pace of 50 rpm throughout the test. Heart rate was monitored and recorded throughout the test. Stable heart rates from two consecutive stages between 110 bpm and 85% age-predicted maximal heart rate were used to predict VO_2max_.

Movement behaviors

Participants’ movement behaviors were measured for two weeks (14 consecutive days) using a triaxial accelerometer (wGT3X-BT, ActiGraph, Pensacola, FL, USA). Participants were instructed to wear the accelerometer on the non-dominant hip, which is a position used previously to measure physical activity [49]. Participants were asked to wear the accelerometer throughout all waking hours except when participating in water-based activities (e.g., swimming and bathing). All participants were provided with instructions concerning the charge and care for the device.

The ActiLife software (Version 6.13.4) was used to initialize the accelerometers and process the data using a midnight–midnight 24 h format. The device was initialized to commence at 12:00 a.m. on Day 1 and to terminate at 12:00 a.m. on Day 14. In this study, raw triaxial acceleration data was collected at a sampling rate of 30 Hz and processed at 10-s epochs with a low frequency extension applied to capture lower magnitude activities [50]. Sleep epochs were identified using the sleep log and then removed. Additionally, in this study, the non-wear-time validation tool available in ActiLife software was applied. The software provided two defaults (Troiano 2007 and Choi 2011) with an allowance for customizing. We conducted a randomized subgroup trial (*n* = 5). After calculating the non-wear period using both defaults, the criteria of Choi 2011 were selected following checks from participants. The default included a minimum length of 90 min for a consecutive 0 counts with 2 min of spike tolerance in an up/down stream small-window length of 30 min [51]. After removing sleep and non-wear periods, as well as invalid data, the remaining data were used to identify the valid control week. This was defined as >4 of 7 days, with >10 h per day during waking time. The valid exercise week was defined as 7 days, with >10 h per day of waking time. Data including both valid control and exercise weeks were used in the final analysis.

The time spent participating in sedentary behaviors and PA at different intensities was calculated by using the Freedson Adult algorithm. The Freedson cut-off points were applied with sedentary defined as <100 counts per minute (cpm), LPA as 100–1951 cpm, MPA as 1952–5724 cpm, VPA as 5725–9498 cpm, and MVPA as >1951 cpm. TPA was identified as the average daily vector magnitude cpm [52]. A prolonged sedentary bout was identified as a minimum of consecutive 30 min in which <100 cpm were recorded [53].


**Exercise exertion**


Borg’s RPE was used to assess participants’ exercise exertion. The 15-point scale ranged from 6 to 20, with 6 indicting no exertion and 20 indicating maximal exertion.


**Exercise protocol**


Participants were required to perform the exercise intervention on Day 8, Day 10, and Day 12 of the designated two-week period. If participants were unable to engage in a scheduled exercise, the exercise was performed on the next day and supervised by the same researcher. On the first visit, participants were instructed to follow the “Timer Plus” App and a familiarization trial was conducted to acquaint participants with exercise protocols.

All exercise sessions included a 10-min low-to-moderate warm-up, a 4-min maximal work-out, and a 5-min cool-down and stretching. The Tabata style HIIE protocol included 4 movements (jumping jacks, high knees, squat jumps, and mountain climbers in sequence) with subjects using their own body weight based on the Tabata training recommendations [54]. During the 20 s exercise period, participants were encouraged to work maximally and to repeat the movements as many times as possible, and then rest for 10 s. There were 8 bouts in each session, with 4 movements completed in sequence that were repeated.

Participants were verbally encouraged to move using maximal efforts. To assess exercise fidelity, a chest strap heart rate monitor (Polar H10, Polar, Malaysia) was applied during each session. Monitors were placed close to the heart and attached by a band to the chest using non-slip silicone dots and a buckle. Heart rate data per second were recorded and processed using the Polar Flow. Maximal heart rate (HR_max_) was calculated using the age-predicted equation (i.e., 220-age), and according to the Tabata protocol, 90% of HR_max_ was required during the 6th bout.

### 2.4. Statistical Analysis

Sample size was estimated by G * Power (version 3.1.9.7) (Heinrich Heine University, Dusseldorf, Germany) using a priori based on the difference between paired means. The effect size was set at 0.80 and alpha was set at 0.05. After calculation, 15 participants were required to achieve 80% power.

Descriptive data were summarized as mean ± SD. Normality was checked using the Shapiro–Wilk test. The correlation was examined using the Pearson’s product moment correlation coefficient. For weekly basis analysis, a paired t-test was used to compare the mean differences of sedentary behaviors (ST, PST, BPST), physical activities (LPA, MPA, VPA, MVPA, and TPA), and sleep between the control week and exercise week. For an additional daily analysis, a repeated measures analysis of variance with the Bonferroni post hoc test was performed to analyze any differences in sedentary behaviors, physical activities, and sleep during the control week, exercise day, and the following day. Linear regression modeling was used to assess the effects of VO_2max_, exercise exertion, and HR_mean_ on changes in movement behaviors. SPSS for windows, version 23.0 (Chicago, IL, USA) was used for statistical analysis, and the significance level was set as *p* < 0.05.

## 3. Results

A total of 29 participants were involved and finally completed all sessions during the three weeks, with all participants attending the exercise on the scheduled day 21 (72.4%). Participants met the inclusion criteria for final analysis, with 19 (90.5%) recording all 14 valid days, and 2 (9.5%) recording 13 valid days (one invalid day on baseline week). The mean wear day was 6.9 ± 0.3 days during the control week and 7.0 ± 0.00 days during the exercise week. The mean daily wear time was 23.1 ± 1.4 and 23.3 ± 0.5 h for the control and exercise week respectively. Baseline characteristics are presented in Table 1. Correlation data for movement behaviors at baseline are detailed in Appendix A.

Weekly basis analysis

Daily LPA significantly decreased from 269.4 ± 55.5 min per day (min/d) during control to 246.0 ± 54.5 min/d during exercise week (*p* < 0.05). MPA, VPA, MVPA, and TPA showed statistically significant increases during the exercise week (*p* < 0.001) and the mean changes were 11.5 ± 12.6, 2.6 ± 2.7, and 14.1 ± 13.1 min/d, and 119.2 ± 70.9, cpm respectively. We found the largest increase in VPA, with a 35.2 ± 40.6% increase during the exercise week.

ST increased 4.4 ± 6.0% during the exercise week, from 446.7 ± 98.9 to 464.1 ± 95.5 min/d (*p* < 0.01). The time spent in prolonged sedentary activities was significantly decreased from 160.4 ± 54.4 to 148.1 ± 50.1 min/d, with a percentage decrease of 5.1 ± 20.6% (*p* < 0.05). No significant differences were found in BPST (*p* > 0.05).

Sleep durations were significantly decreased from 9.0 ± 1.1 h per day (h/d) to 8.5 ± 0.7 h/d, with a percentage decrease of 5.0 ± 7.5% (*p* < 0.05).

See weekly changes in Figure 2 and Table 2.

Daily basis analysis

LPA decreased significantly on the exercise day from 269.4 ± 55.5 to 245.9 ± 50.8 min/d and at the day after exercise, it significantly increased to 250.9 ± 53.1 min/d but was still significantly below the control (*p* < 0.05) (Figure 3a1). MPA, VPA, and MVPA increased significantly on the exercise day from 148.3 ± 47.4 to 170.2 ± 46.9 min/d, 11.1 ± 7.2 to 14.0 ± 7.5 min/d, and from 159.5 ± 52.0 to 184.2 ± 51.6 min/d, respectively (*p* < 0.05). However, on the following day, they decreased significantly and returned to control levels (Figure 3a2–a4). TPA was 1605.8 ± 401.4 cpm on exercise day, which was significantly higher than the control (1411.5 ± 381.5 cpm), and significantly decreased the next day to 1472.8 ± 366.4 cpm (*p* < 0.05), staying above the control (*p* < 0.05) (Figure 3b).

We found that ST increased continuously after performing exercises, and on the day after exercise it demonstrated the biggest increase (446.7 ± 98.9, 454.6 ± 101.3, and 473.6 ± 103.3 min/d for the control, exercise day, and next day, respectively, *p* < 0.05, Figure 3c1). However, time spent on PST declined significantly on the exercise day from 160.4 ± 54.4 to 152.0 ± 47.5 min/d (*p* < 0.05) and then remained unchanged the next day (Figure 3c2).

Sleep duration revealed significant decreases on the exercise day from 9.0 ± 1.1 to 8.6 ± 0.8 h/d (*p* < 0.05) and remained unchanged the day after exercise (8.5 ± 0.8 h/d, *p* > 0.05, Figure 3d).

See percentage daily changes in Figure 4 and details in Table 3.

Exercise fidelity and exertion

The mean heart rate during exercises was 82.4 ± 1.9% of HR_max_ (ranging from 79.4% to 85.8%), and the mean peak heart rate was 92.3 ± 3.1% of HR_max_, with the lowest being 86.3% and the highest being 97.0%. Seventeen (81.0%) participants met the high intensity requirements of 90% HR_max._ The average peak heart rate achieved during Session 1 to Session 3 was 92.3 ± 3.0, 92.4 ± 2.8, and 92.4 ± 3.8% respectively, with no between-differences. The reported score for Borg’s RPE was 15.7 ± 0.5 during exercise, ranging from 14.7 to 16.7.

Linear regression modeling

VO_2max_, Borg’s RPE score, and HR_mean_ were included in the linear regression model to explore the potential variables associated with changes in movement behaviors. Only the changes in ST from the control week to the exercise week were significantly associated with VO_2max_ (b = −0.36, 95% CI: −0.72, −0.01; *p* = 0.047) and HR_mean_ during exercises (b = 0.54, 95% CI: 0.19, 0.88; *p* = 0.004), indicating that better aerobic capacity was associated with a smaller increase in ST, and that higher exercise intensity led to more increases in ST.

## 4. Discussion

This study was the first to investigate the compensatory effects of HIIE among inactive female university students across an entire 24-h day. In the present study, we observed a compensatory increase in ST, with the most increases occurring on the day after exercise, with decreases in LPA. On the contrary, MPA, VPA, MVPA, and TPA increased on exercise day. Sleep duration decreased after HIIE.

Findings from the present study were partially supported by an experimental study conducted in adolescents by Paravidino et al., (2017) [29]. There was an increase in subsequent sedentary time after both 55 min of moderate (64–76%HRmax) and vigorous (77–95%HRmax) exercise sessions. Similar behavior compensations for acute exercise by increasing ST and PST and decreasing LPA were also reported among older adults [55]. Although our results indicated that HIIE induced compensation for the increasing ST and decreased LPA occurring in inactive young women, the simultaneous decreases in MVPA following HIIE reported by previous studies were not observed in the present study.

Inconsistent to the “Activitystat” hypothesis, MPA, VPA, MVPA, and TPA were increased rather than reduced following HIIE in the current study, which was consistent with several other studies. Cooper et al., (2003), Goodman et al., (2011), and Long et al., (2013) investigated the effects of participating in PA among children and found that PA interventions may increase activity overall, indicating the occurrence of activity synergy in youth [21,22,23]. We further found that low-volume HIIE-induced activity synergy was transient, as it was observed only on exercise day and it returned to baseline levels the next day. However, an observational study conducted by Baggett et al., (2010) showed a positive correlation between daily MVPA and MVPA on the following days among female adolescents [56]. We hypothesized that in the present study, higher levels of PA on the exercise day were mostly accumulated as the result of increased exercise. Meanwhile, the transient effect of activity synergy could be potentially explained by the short workout bout and the low exercise volume in the current study. This suggestion had been highlighted in a systematic review of randomized controlled trials [57], in which the session duration was revealed to be significantly positively associated with compensatory behaviors.

Exercise intensity was another factor potentially influencing subsequent movement behaviors. Several studies assessed the changes of behaviors after engaging in exercises with different protocols. One study compared the compensatory effects of a single bout of moderate intensity training, high intensity exercise, and sprint training, and reported greater declines in PA following high intensity exercise and sprint training [55]. However, in a 22-week intervention, no compensatory changes were reported in four exercise groups (strength, endurance, combined, and PA recommendations) [58]. Our study supported the findings that exercises including higher intensities were more likely to induce movement behavior compensations, since the HRmean was significantly positively associated with an increase in ST after HIIE. We also found females with a better aerobic capacity had smaller increases in ST after HIIE. It was worth also noting that in our study, there was no correlation between perceived exertion and increases in ST. Participants with a higher Borg’s RPE score did not subsequently spend more time sitting, sleeping, or engaging in other activities.

In contrast, several studies reported no changes in non-exercise behaviors after exercise interventions. De Moura et al., (2015) noted that compensatory effects were not observed following an 8-week moderate intensity aerobic exercise [59]. Similar findings were indicated after an 8-month training program with a moderate to high intensity [20]. Furthermore, in the study conducted by Church et al., (2007), non-exercise physical activities were assessed following different doses of exercises and, during 6 months of intervention, neither behavior compensation nor synergy were observed [60]. One of the potential explanations of these findings was that the longer intervention length was evidenced to be favorably associated with non-exercise-based physical activities [57].

When investigating compensatory activities, most studies have focused on changes of MVPA and ST. As a growing number of studies have indicated the health promotion effect of LPA, we had to take LPA into consideration simultaneously when evaluating changes on LPA. Our study revealed a decrease in LPA after HIIE in both weekly and daily analyses. This finding was consistent with a study conducted by Ridgers et al., (2014), in which additional time spent in MVPA was associated with less time spent in LPA on subsequent days [61]. However, in an observational study, daily LPA was positively associated with MVPA [56]. Similarly, a significantly higher LPA was revealed in the school day with physical education classes when compared to a day without physical education classes [62]. These controversial results might be caused by the different cut-off points, epochs, and placements applied to the accelerometer measurements, which result in differences in identifying between LPA and MVPA.

Additionally, it was interesting that total time spent in PA, including LPA, MPA and VPA, was unchanged, while TPA, measured by the vector magnitude cpm, increased significantly. Correlation analysis also indicated a significantly positive association between TPA with MPA, and VPA and MVPA but not LPA. We tested the changes of PA on the exercise day when controlling for interventions by removing exercise periods and found that time spent on MPA still increased significantly, while VPA showed a decrease. In addition to the increased non-exercise MPA, the PST accumulated in our study was significantly reduced following HIIE. Sedentary behavior that is accumulated in a prolonged manner has been independently correlated to an increased risk of cardiometabolic diseases [32]. At this point, we believe that participating in such low-volume HIIE would make participants more active on the same day. Even though the 24 h was limited, it was an effective way to increase overall PA by replacing lower intensity PA with higher intensity.

Sleep duration was significantly decreased following HIIE. This was supported by a day-to-day study that indicated there was a negative association between steps and sleep duration [63]. However, some studies suggested that exercises were an effective strategy to improve sleep duration. Mendelson et al., (2016) conducted a 12-week monitored exercise intervention to improve sleep duration and found it was effective, with sleep duration increasing from 6.7 to 7.4 h [64]. Another study examined the effects of exercises at different intensities on sleep duration and indicated a short-term increase on sleep duration only after vigorous intensity exercise [65]. Kakinami et al., (2017) reported no association between PA intensity and sleep quantity [66]. The contradiction in results might be due to the different characteristics of sleep at baseline. Participants in our study revealed a longer sleep duration close to the maximum of the recommended level. Both inadequate and longer sleep duration have negative effects on health in women [67]. Even though the time spent on sleep decreased following HIIE, it was not a negative effect, but rather can be recognized as an improvement in sleep quality resulting from increased activity.

The HIIE protocol revealed an acceptable fidelity with high intensity exercise achieved by most participants. It was interesting that Borg’s RPE score was not associated with changes in any movement behaviors. We estimated heart rate with the RPE score, using the equation recommended by Scherr et al., (2013): HR = 69.34 + 6.23 × RPE [68]. The estimated HR was significantly lower than the objectively measured one, suggesting that high intensity exercises might be perceived as being less strenuous when performing Tabata style exercises. Lack of time and exercise exertion was reported as the most cited barriers to exercise among females; therefore, a Tabata-style HIIE represents a viable alternative to improve PA among inactive young females

Our study has several strengths including (1) the daily average length of accelerometry data per participant being more than 23 h. This is extensively higher than previous studies and consisted of evaluating 24-h movement behaviors, therefore increasing the validity of findings; (2) the inclusion of daily and weekly analysis made contributions to the comparation between acute and short-term effects; (3) the discrepancy between subjectively and objectively measured intensities indicated the perceived exertion during the Tabata-style HIIE.

Several limitations also need to be mentioned. Firstly, the small sample size and participants’ characteristics limit the extrapolation of our findings. Participants included in the present study were young, female, of normal weight, and inactive. Secondly, energy expenditure was not assessed due to the invalid estimation by ActiGraph [69]. Furthermore, although the age-predicted maximal heart rate is a quick and easy estimation, it is suggested to overestimate it in young adults. Using a specialized equation would be more appropriate as age and gender might influence the accuracy of the estimation. Finally, as this trial was a pilot study, the long-term effects of HIIE on the changes of movement behaviors needs to be investigated further.

## 5. Conclusions

Compensatory changes in movement behaviors were recognized as being potential explanations for the varied effects on health outcomes following HIIE. The results highlighted the existence of compensatory increases in ST in response to a low-volume HIIE in inactive young females, and the magnitude of increase in ST was associated with exercise intensity and aerobic capacity. Although LPA decreased, TPA was increased following HIIE, and MPA, VPA, MVPA temporary improved on the exercise day. Time spent in prolonged sedentary behaviors and sleep was reduced. Overall, in the short term, participating in a low-volume HIIE made participants more active since the observation of an increase in ST may displace prolonged sitting and an overlong sleep period. Higher-intensity PA may displace lower levels of PA. Although low-volume HIIE may contribute to an increase in physical activity in a short term, further investigation is needed to develop an understanding of the long-term effects of changes in movement behaviors following HIIE.

## Figures and Tables

**Figure 1 ijerph-19-07177-f001:**
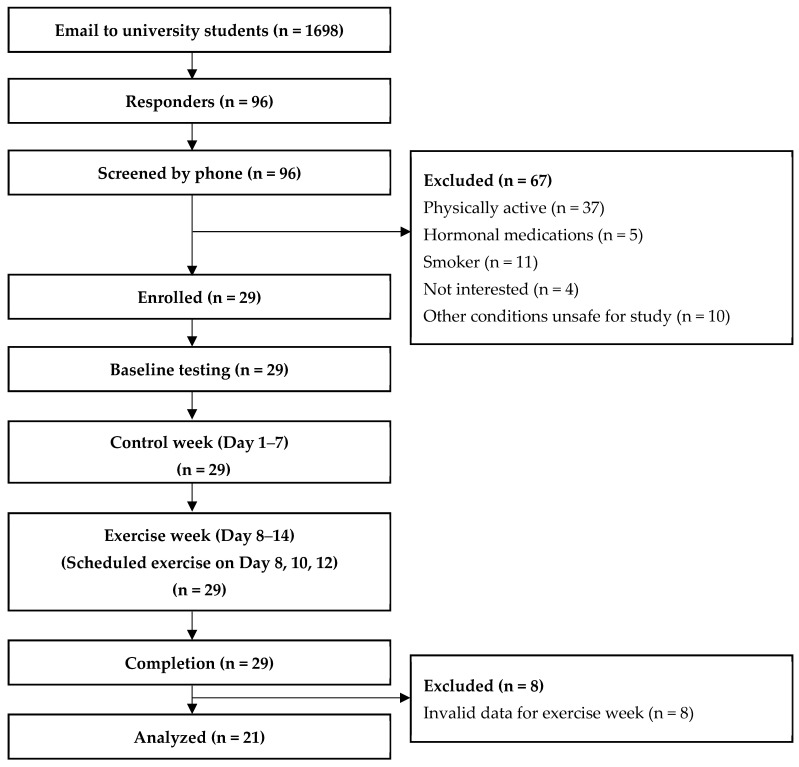
Flow diagram of sample and study timeline.

**Figure 2 ijerph-19-07177-f002:**
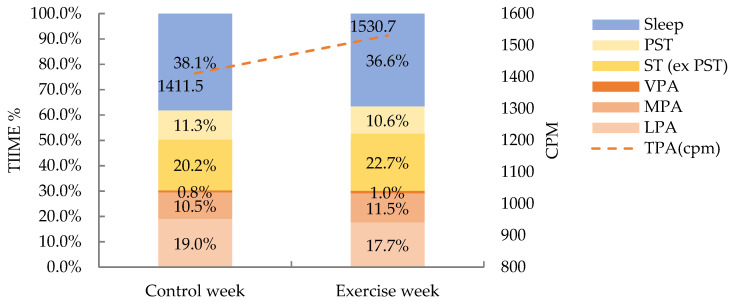
The changes of movement behaviors between control week and exercise week Note: cpm, counts per minute; LPA, light intensity physical activity; MPA, moderate intensity physical activity; MVPA, moderate to vigorous intensity physical activity; PST, prolonged sedentary time; ST (ex PST), sedentary time except PST; TPA, total physical activity; VPA, vigorous intensity physical activity.

**Figure 3 ijerph-19-07177-f003:**
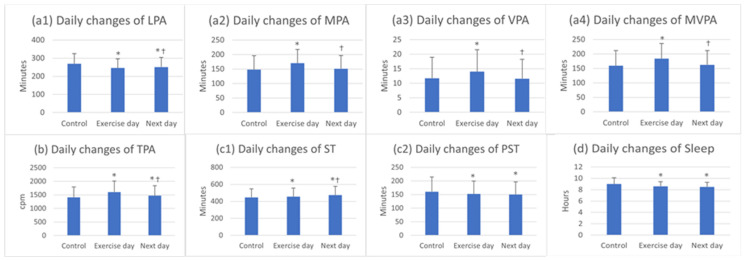
Daily changes of movement behaviors. Note: cpm, counts per minute; LPA, light intensity physical activity; MPA, moderate intensity physical activity; MVPA, moderate to vigorous intensity physical activity; PST, prolonged sedentary time; ST, sedentary time; TPA, total physical activity; VPA, vigorous intensity physical activity. *, statistically significant difference from control *p* < 0.05; ^†^, statistically significant difference from exercise day *p* < 0.05.

**Figure 4 ijerph-19-07177-f004:**
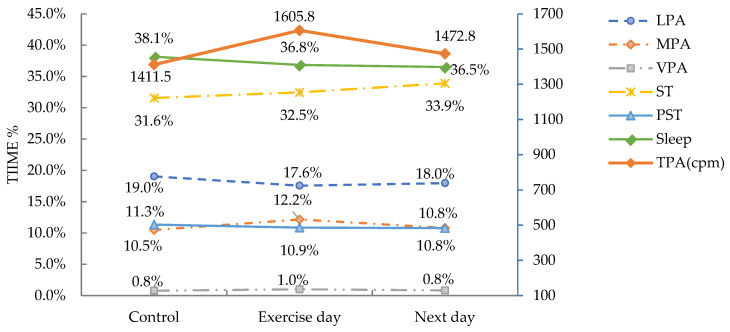
Daily changes of movement behaviors. Note: cpm, counts per minute; LPA, light intensity physical activity; MPA, moderate intensity physical activity; MVPA, moderate to vigorous intensity physical activity; PST, prolonged sedentary time; ST, sedentary time; TPA, total physical activity; VPA, vigorous intensity physical activity.

**Table 1 ijerph-19-07177-t001:** Characteristics of participants at baseline (*n* = 21).

Variables	Mean ± SD
Age (years)	25.4 ± 1.0
Weight (kg)	60.9 ± 4.9
Height (m)	165.4 ± 3.9
Body mass index (kg/m^2^)	22.3 ± 1.9
VO_2max_ (mL/kg/min)	36.8 ± 4.3
Accelerometry (days)	13.9 ± 0.3
Wear time (h/d)	23.6 ± 0.2
Movement behaviors:	
LPA (min/d)	269.4 ± 55.5
MPA (min/d)	148.3 ± 47.4
VPA (min/d)	11.1 ± 7.2
MVPA (min/d)	159.5 ± 52.0
TPA (cpm)	1411.5 ± 381.5
ST (min/d)	446.7 ± 98.9
PST (min/d)	160.4 ± 54.4
Sleep (h/d)	9.0 ± 1.1

Note: bpm, beat per minute; cpm, counts per minute; h/d, hours per day; LPA, light intensity physical activity; min/d, minutes per day; MPA, moderate intensity physical activity; MVPA, moderate to vigorous intensity physical activity; PST, prolonged sedentary time; ST, sedentary time; TPA, total physical activity; VPA, vigorous intensity physical activity.

**Table 2 ijerph-19-07177-t002:** Weekly changes of movement behaviors.

Variables	Control Week	Exercise Week	Mean Change	95% CI		% Change	*p* Value	Effect Size (Cohens’d)
LPA (min/d)	269.4 ± 55.5	246.0 ± 54.5	−23.4 ± 44.3 *	−43.5	−3.2	−7.3 ± 16.7%	*p* < 0.05	−0.53
MPA (min/d)	148.3 ± 47.4	159.8 ± 44.8	11.5 ± 12.6 ***	5.7	17.2	9.9 ± 11.7%	*p* < 0.001	0.91
VPA (min/d)	11.1 ± 7.2	13.8 ± 7.6	2.6 ± 2.7 ***	1.4	3.8	35.2 ± 40.6%	*p* < 0.001	0.96
MVPA (min/d)	159.5 ± 52.0	173.6 ± 49.1	14.1 ± 13.1 ***	8.2	20.1	11.1 ± 11.6%	*p* < 0.001	1.06
TPA (cpm)	1411.5 ± 381.5	1530.7 ± 384.3	119.2 ± 70.9 ***	86.9	151.5	9.1 ± 5.6%	*p* < 0.001	1.68
ST (min/d)	446.7 ± 98.9	464.1 ± 95.5	17.5 ± 25.9 **	5.7	29.3	4.4 ± 6.0%	*p* < 0.01	0.68
PST (min/d)	160.4 ± 54.4	148.1 ± 50.1	−12.3 ± 24.7 *	−23.5	−1.0	−5.1 ± 20.6%	*p* < 0.05	−0.49
Sleep (h/d)	9.0 ± 1.1	8.5 ± 0.7	−0.5 ± 0.7 **	−0.8	−0.2	−5.0 ± 7.5%	*p* < 0.01	−0.71

Note: cpm, counts per minute; h/d, hours per day; LPA, light intensity physical activity; min/d, minutes per day; MPA, moderate intensity physical activity; MVPA, moderate to vigorous intensity physical activity; PST, prolonged sedentary time; ST, sedentary time; TPA, total physical activity; VPA, vigorous intensity physical activity. * *p* < 0.05; ** *p* < 0.01; *** *p* < 0.001.

**Table 3 ijerph-19-07177-t003:** Daily changes of movement behaviors.

Variables	Control	Exercise Day	Next Day	*p*-Value	*η* ^2^
LPA (min/d)	269.4 ± 55.5	245.9 ± 50.8 *	250.9 ± 53.1 *^†^	<0.001	0.615
MPA (min/d)	148.3 ± 47.4	170.2 ± 46.9 *	151.0 ± 45.5 ^†^	<0.001	0.784
VPA (min/d)	11.1 ± 7.2	14.0 ± 7.5 *	11.6 ± 6.6 ^†^	<0.001	0.735
MVPA (min/d)	159.5 ± 52.0	184.2 ± 51.6 *	162.6 ± 48.9 ^†^	<0.001	0.816
TPA (cpm)	1411.5 ± 381.5	1605.8 ± 401.4 *	1472.8 ± 366.4 *^†^	<0.001	0.784
ST (min/d)	446.7 ± 98.9	454.6 ± 101.3 *	473.6 ± 103.3 *^†^	<0.001	0.462
PST (min/d)	160.4 ± 54.4	152 ± 47.5 *	150.5 ± 45.8 *	<0.001	0.35
Sleep (h/d)	9.0 ± 1.1	8.6 ± 0.8 *	8.5 ± 0.8 *	<0.001	0.419

Note: cpm, counts per minute; h/d, hours per day; LPA, light intensity physical activity; min/d, minutes per day; MPA, moderate intensity physical activity; MVPA, moderate to vigorous intensity physical activity; PST, prolonged sedentary time; ST, sedentary time; TPA, total physical activity; VPA, vigorous intensity physical activity. *, statistically significant difference from control *p* < 0.05; ^†^, statistically significant difference from exercise day *p* < 0.05.

## Data Availability

The data presented in this study are available on request from the corresponding author. The data are not publicly available due to student privacy.

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
