# Peer review of "Effects of Low-Volume High-Intensity Interval Exercise on 24 h Movement Behaviors in Inactive Female University Students"

_ijerph, 2022, doi:10.3390/ijerph19127177_

Round 1

Reviewer 1 Report

Thank you very much for letting me review this manuscript.

In my opinion it is a manuscript that reflects a topic of current interest, sedentary and female, and has all the aspects to be considered publishable.

Just one point: having so much data and tables of results, I would be inclined to present the results more graphically and paying a lot of attention to the visualisation of the data, I think it would make the manuscript much more appealing to the reader.

Congratulations on the work done

Author Response

Dear Reviewer,

All the line numbers in my last response corresponded to the manuscript displayed in all-visible “Track Changes” in Microsoft Office Word. In consideration of the fact that you may have received a PDF version with all track changes invisible, we have changed all line numbers in this new response. Sorry for the inconvenience and hope to meet with the final approval.

Comments:

  1. Just one point: having so much data and tables of results, I would be inclined to present the results more graphically and paying a lot of attention to the visualisation of the data, I think it would make the manuscript much more appealing to the reader.

Response: Thank you very much for your comments. We added Figure 3 to make the daily data visible. Furthermore, we simplified the table to make it more clear now and we also added the value of effect size, as in line 303-327.

Once again, thank you very much for your suggestions and comments, and we feel highly honored by your support

Reviewer 2 Report

Authors should describe the training performed.

The authors must justify why they used the 220-age equation for training control, as this equation is not validated.

Author Response

Dear Reviewer,

All the line numbers in my last response corresponded to the manuscript displayed in all-visible “Track Changes” in Microsoft Office Word. In consideration of the fact that you may have received a PDF version with all track changes invisible, we have changed all line numbers in this new response. Sorry for the inconvenience and hope to meet with the final approval.

Comments:

  1. Authors should describe the training performed.

Response: Thank you very much for your comments. We added the description of exercise attendance and details about the wear time in the section of results as in Line 234-239. Additionally, we added the description of peak heart rate achieved during exercise sessions as in Line 336-338.

  1. The authors must justify why they used the 220-age equation for training control, as this equation is not validated.

Response: Thank you very much for your comments. Although this equation is limited in predicting the accurate maximal heart rate, it is a quick and easy estimation. Since this equation is reported to overestimate in young adults, the maximal heart rate for young student is probably overestimated in our study and it seems that the actual exercise intensity achieved is higher than observed. Our second aim is to examine whether such high intensity exercise can be achieved and accepted by previous inactive individual, and in this regard, at least, the answer is Yes! Furthermore, although the limited accuracy it documented, the equation is still recommended in clinical settings and published in resources by well-established organizations in the field [Fletcher et al., 2013]. Finally, we described it as a limitation in our study as in Line 458-461.

Once again, thank you very much for your suggestions and comments, and we feel highly honored by your support

Reviewer 3 Report

The title and abstract are quite clear and precise. The steps taken can be understood.

Introduction

The introduction is well written and technically well structured. It is even interesting and arouses the need to quickly find out what will be written next.

Method

This chapter is too long for my taste. There may be too much unnecessary data. However, it has all the necessary elements, and there are no omissions, so I can’t request corrections.

Results

The results are clearly shown.

Discussion

The discussion is too long and boring, but it also has all the necessary elements.

Conclusion

The conclusion is clearly and precisely written.

Final statement:

This manuscript has all the technical elements. It cannot be said that it does not provide interesting data, however it is not adequate for a journal of IJERPH rank in my opinion. I do not see how these results can be put to good use.

Unfortunately, I have to recommend that this manuscript be rejected or recommended for a lower-ranking journal.

However, I would leave the possibility to the Editor-in-Chief to do the opposite, because the manuscript is technically well prepared and the data it provides are interesting.

Author Response

Dear Reviewer,

All the line numbers in my last response corresponded to the manuscript displayed in all-visible “Track Changes” in Microsoft Office Word. In consideration of the fact that you may have received a PDF version with all track changes invisible, we have changed all line numbers in this new response. Sorry for the inconvenience and hope to meet with the final approval.

Response: Thank you very much for your comments. We did several revisions based on your comments. Firstly, we remove the variable of breaks in sedentary time, as it indeed seems an unnecessary data and add more description about peak heart rate since it is an important variable to examine the intensity achieved during exercise sessions. Secondly, we simplify the table and make it more visible in figures. Finally, we add the description of research gap regarding to high intensity interval exercise (HIIE) as in Line 55-81. Although HIIE is popular among young females today, there is no consistent evidence that it is favorable associated with fat and body mass reduction. Furthermore, its mechanism of reducing fat is still unclear because basically, when the exercise intensity reaches 85% VO2max and more, the energy metabolism is almost supplied by sugar and few fats is consumed. Therefore, we hypothesize that other factors may affect the effect of HIIE on fat reduction. One of the potential explanations is that individuals become more physically active in the daily life after engaging in HIIE, however, it need further investigation. Overall, to our knowledge, it is the first study to examine movement behaviors within a 24-hour framework after HIIE in young female adults, and we hope that our findings can provide useful and important information to the physical activity prescription and health promotion program. However, further research should seek evidence on its long-term effects on lifestyle and health and identify optimal ways to deliver such vigorous exercise to inactive individuals.

Once again, thank you very much for your suggestions and comments, and we feel highly honored by your support!

Reviewer 4 Report

Introduction:

Overall, there is a good guiding thread for the introduction, which facilitates the description of the variables under study. However, some points should be tackled:

Comment 1. Page 2, line 51: lease, add a reference in MDPI style for King et al. (2007).

Comment 2. Page 2, lines 56 and 57: Please, add a body de research for the argumentation of this sentence: “Although the effects of HIIE on improving health outcomes has been investigated by 56 numerous studies,”

Comment 3.  Page 3, lines 57 and 58: Please, it is needed to provide additional information about the results from the few studies on this topic.

Comment 4. Which importance has the study of 24-hour movement behaviours? Which does this research add to the existing basis of studies?

Comment 5. Please, it is needed to provide rationale for the selection of female participants throughout this section. Indeed, the topic of different MVPA levels between male and female students might be a starting point for the study of female participants.

 Method:

Although the method section was well described, there are some minor points that need be addressed:

Comment 1: Participants: Please, it is needed to detail the type of sampling strategy followed for recruitment and selection of participants.

Comment 2. Procedure: Please, provide information about the type of quasi-experimental design. To the best of my knowledge, the absence of a control group would be indicative of a pre-experimental design.

Comment 3: Data analysis: Please, it is required to provide further information about the test statistic used for repeated measures analysis of variance, and the choice of the Bonferroni’s correction instead of another ones (Field, 2017). Similarly, it is needed to reported effect size evidence for the different tests run. For instance: Cohen’s d coefficient for Student  t test and eta partial squared for repeated measures analysis of variance (Field, 2017). On the other hand, there is a need to provide evidence in support of the absence of multicollinearity among independent variables for linear regression model.

Results

In my viewpoint, the results were shown in a brief and accurate manner. Congrats.

 Discussion

In my opinion, discussion is well approached by arguing and contrasting the obtained results with those from previous research.

Author Response

Dear Reviewer,

All the line numbers in my last response corresponded to the manuscript displayed in all-visible “Track Changes” in Microsoft Office Word. In consideration of the fact that you may have received a PDF version with all track changes invisible, we have changed all line numbers in this new response. Sorry for the inconvenience and hope to meet with the final approval.

Comments:

  1. Page 2, line 51: lease, add a reference in MDPI style for King et al. (2007).

Response: Thank you very much for your comments. We added the reference in MDPI style as in Line 50.

  1. Page 2, lines 56 and 57: Please, add a body de research for the argumentation of this sentence: “Although the effects of HIIE on improving health outcomes has been investigated by 56 numerous studies,”

Response: Thank you very much for your comments. We added 4 relevant references as in Line 59. These are Sultana et al., 2019, Astorina et al., 2012, De Revere et al., 2021, Gibala et al., 2018.

  1. Page 3, lines 57 and 58: Please, it is needed to provide additional information about the results from the few studies on this topic.

Response: Thank you very much for your comments. We added findings from previous studies as in Line 60-67. A previous study evaluated the within-day changes of sedentary behaviors after vigorous physical activity and reported an increase in subsequent sedentary time [Skovgaard et al.,2019]. Another crossover study on overweight boys examined the behavioral changes after a single bout of moderate or vigorous exercise. The author indicated a slightly higher increase in sedentary time after vigorous exercise than moderate in the following 4 days, and furthermore, the time spent in vigorous intensity physical activity (VPA) decreased greatly after vigorous exercise [Paravidino et al., 2017]. While in the study conducted in adult, time spent in sedentary activities was significantly reduced during HIIE [Nugent et al., 2018].

  1. Which importance has the study of 24-hour movement behaviours? Which does this research add to the existing basis of studies?

Response: Thank you very much for your comments. Because 24 hours a day is fixed, a change in one behavior is associated with changes in other behaviors. For example, if ones spend more time on sedentary behaviors in a day, they spend less time on physical activity, sleep, or other behaviors. Since sedentary behavior, physical activity, and sleep are all associated with health outcomes, any changes in one behavior can induce changes on others and further affect the health outcomes. We added the description of the important of 24-hour framework as in Line 70-73.

On the other hand, many studies examined the effects on high intensity interval exercises on health outcomes and there is moderate evidence that high intensity interval exercise can improve cardiorespiratory fitness and cardiometabolic health. However, its effect on movement behaviors especially 24-hour movement behaviors is still unclear. Therefore, our study can provide useful and important information to the high intensity interval exercise in the use of physical activity prescription and health promotion program as all movement behaviors are associated with health outcomes as well.

  1. Please, it is needed to provide rationale for the selection of female participants throughout this section. Indeed, the topic of different MVPA levels between male and female students might be a starting point for the study of female participants.

Response: Thank you very much for your comments. We added the description about why we choose young females as in Line 55-57. Recently, the prevalence and popularity of HIIE among young adults have made it an exciting type of exercise and health promotion intervention, especially among females as it is promoted as a time-efficient and effective strategy to reduce fat [Maillard et al., 2018]. It seems that such type of exercise is more attractive for females and moreover, the physical activity level is lower in females compared in males. Therefore, it fits the aim of our study to explore changes on movement behaviors after engaging in high intensity interval exercises.

  1. Participants: Please, it is needed to detail the type of sampling strategy followed for recruitment and selection of participants.

Response: Thank you very much for your comments. We deleted the previous Table 1 and added a flow diagram for participants selection and study timeline as in the Table 1 revised.

  1. Procedure: Please, provide information about the type of quasi-experimental design. To the best of my knowledge, the absence of a control group would be indicative of a pre-experimental design.

Response: Thank you very much for your comments. Due to several practical reasons such as the small sample size, the shortage of accelerometers, and the high difference on exercises time, we are unable to set a randomized equivalent control group. Therefore, we determined to use a quasi-experimental design in which all participants served as their own controls.

  1. Data analysis: Please, it is required to provide further information about the test statistic used for repeated measures analysis of variance, and the choice of the Bonferroni’s correction instead of another ones (Field, 2017). Similarly, it is needed to reported effect size evidence for the different tests run. For instance: Cohen’s d coefficient for Student  t test and eta partial squared for repeated measures analysis of variance (Field, 2017). On the other hand, there is a need to provide evidence in support of the absence of multicollinearity among independent variables for linear regression model.

Response: Thank you very much for your comments. We added the effect size for both weekly and daily analysis as in Table 2 and Table 3. We choose the Bonferroni’s correction because it well controls the Type I error and makes the result more precise. Furthermore, there are not many comparisons tested in our study and the use of Bonferroni’s correction will not be too conservative. Finally, we tested the multicollinearity for all variables in the regression model and there is no multicollinearity. The value of variance inflation factor for VO2max, mean heart rate, and the Borg’s RPE score is 1.586, 1.457 and 1.120 respectively.

Once again, thank you very much for your suggestions and comments, and we feel highly honored by your support

Round 2

Reviewer 2 Report

The authors made the necessary adjustments to the article.

Congratulations

Reviewer 4 Report

In my viepoint, the authors made a great labour in responding to every comment proposed. I full satisfied with the current version of the manuscript.